# Detection of Bagaza Virus in Europe: A Scoping Review

**DOI:** 10.3390/vetsci12020113

**Published:** 2025-02-02

**Authors:** Filipa Loureiro, João R. Mesquita, Luís Cardoso, Ana C. Matos, Manuela Matos, Ana Cláudia Coelho

**Affiliations:** 1Wildlife Rehabilitation Centre (CRAS), Veterinary Teaching Hospital, University of Trás-os-Montes e Alto Douro (UTAD), 5000-801 Vila Real, Portugal; 2Animal and Veterinary Research Centre (CECAV), Associate Laboratory for Animal and Veterinary Sciences (AL4AnimalS), University of Trás-os-Montes e Alto Douro (UTAD), 5000-801 Vila Real, Portugal; lcardoso@utad.pt (L.C.); accoelho@utad.pt (A.C.C.); 3School of Medicine and Biomedical Sciences (ICBAS), Porto University, 4099-022 Porto, Portugal; jrmesquita@icbas.up.pt; 4Epidemiology Research Unit (EPIUnit), Instituto de Saúde Pública da Universidade do Porto, 4050-600 Porto, Portugal; 5Laboratory for Integrative and Translational Research in Population Health (ITR), 4050-600 Porto, Portugal; 6Department of Veterinary Sciences, School of Agrarian and Veterinary Sciences (ECAV), University of Trás-os-Montes e Alto Douro (UTAD), 5000-801 Vila Real, Portugal; 7Research Centre for Natural Resources, Environment and Society (CERNAS), Polytechnic Institute of Castelo Branco, 6001-909 Castelo Branco, Portugal; acmatos@ipcb.pt; 8Quality of Life in the Rural World (Q-RURAL), Polytechnic Institute of Castelo Branco, 6001-909 Castelo Branco, Portugal; 9Centre for the Research and Technology of Agro-Environmental and Biological Sciences (CITAB), University of Trás-os-Montes e Alto Douro (UTAD), 5000-801 Vila Real, Portugal; mmatos@utad.pt

**Keywords:** BAGV, birds, encephalitis, *Orthoflavivirus*, red-legged partridge, vector-borne disease

## Abstract

The flaviviruses (family Flaviviridae) comprise aetiological agents of neurological diseases in animals and humans, which are spread worldwide. The Bagaza virus (BAGV), a member of the genus *Orthoflavivirus*, is a mosquito-borne arbovirus with wild birds as amplifying hosts. BAGV is still not well-documented in human health but has already been identified several times in bird species on the European continent. BAGV has been introduced into Spain in at least three independent outbreaks and has also been detected in Portugal and France. In this scoping review, 114 studies were initially evaluated and 12 were included after applying the exclusion criteria.

## 1. Introduction

The family Flaviviridae is a large group of unsegmented positive-sense single-stranded RNA viruses, with a genome size of approximately 10–11 kb which comprises several important human pathogens, such as the viruses of Dengue (DENV), Yellow Fever, and West Nile (WNV). The RNA genome of a flavivirus virion consists of a single long open reading frame flanked by 5′ and 3′ untranslated regions, which is translated into a polyprotein comprising three structural proteins (C, prM, and E protein) and seven non-structural proteins (NS1, NS2A, NS2B, NS3, NS4A, NS4B, and NS5) [1]. Bagaza virus (BAGV) belongs to the genus *Orthoflavivirus*—the genus name *Flavivirus* was changed in the last couple of years to *Orthoflavivirus* by the International Committee on Taxonomy of Viruses (ICTV)—(Ntaya serocomplex) and is maintained through an epidemiological cycle involving mosquitoes as competent vectors and birds as natural reservoirs and virus amplifiers [2]. The BAGV genome has been found to be very similar to the Israel turkey meningoencephalomyelitis virus (ITV) [3,4], but they are still considered two different agents according to the ICTV [5].

BAGV was first isolated in Bagaza, Central African Republic, in 1966, from a pool of *Culex* spp. mosquitoes [6]. After that, it has also been found in mosquitoes in other countries, namely Mauritania and Senegal [7], Namibia [8] and the United Arab Emirates [9]. In similarity to other *Orthoflavivirus*, like Usutu virus (USUV) [10], it has dispersed to Europe. In this continent, BAGV was detected for the first time in Spain in 2010, and also for the first time in vertebrate hosts [11]. Since then, it has been described in different wild bird species [2,11,12,13,14], besides mosquitoes [15].

It is indisputable that Europe has been experiencing climate changes in recent years, and the situation is becoming unpredictable. Climate change has an influence on the transmission of vector-borne pathogens: it influences the epidemiology of certain diseases, favors the dispersal of vectors, and may contribute to the triggering of some outbreaks, leading to millions of infections [16,17]. In recent decades, there has been an increase in the occurrence and geographical spread of a variety of flavivirus infections transmitted by mosquitoes, which have wild birds as reservoirs. DENV is endemic in more than 100 countries, affecting Southeast Asia and the Western Pacific in particular. USUV and WNV are proven to be emerging diseases in new and unaffected geographical regions, like Europe [18,19]. Environments promoting mosquito breeding and accumulation significantly increase the risk of flavivirus infections. Mosquitoes acquire the virus during a blood meal of an infected host. The virus replicates in the avian amplifying host and is incidentally transmitted to dead-end hosts [18], like humans or horses, in the case of WNV and USUV. BAGV was identified as an emerging and re-emerging pathogen with the potential to cause infections in humans [20]. It was serologically detected in humans [21], but its pathogenicity in people is still unknown. Considering that there are already endemic diseases caused by flavivirus in Europe, namely tick-borne encephalitis and West Nile fever, which can lead to serious neurological conditions [22], care and attention are recommended. BAGV zoonotic potential is still understudied, and surveillance of the virus and associated vectors will allow more effective and rapid control measures, decreasing its negative impact on public and animal health. This study aims to gather all available information on BAGV detection in Europe in the form of a systematic review.

## 2. Materials and Methods

The present review includes articles published before 10th December 2024 in three databases: PubMed, ScienceDirect, and Scopus. Guidelines from The Preferred Reporting Items for Systematic Reviews and Meta-Analysis (PRISMA) were followed for the elaboration of this systematic review [23]. Only studies published, indexed, and peer-reviewed were considered. Language restrictions were included, and only articles written in English were considered. The literature search used the following keywords: (Bagaza or BAGV) AND Europe. After reading the title and the abstract, papers that did not address the detection of BAGV in any European country were excluded. The screening of the databases and the data extraction were performed by two independent investigators (FL and ACC). Differences of opinion over the inclusion of an article were resolved through discussion between the two. The first step was to eliminate duplicate articles (*n* = 19), then only research articles, letters, or short communications were moved forward (*n* = 40 excluded), and finally, exclusion criteria were applied for unrelated research results (*n* = 43). One article in French was found, but it was excluded in the first analysis because it was a review, so it is not mentioned in the flowchart (Figure 1). The application of inclusion and exclusion criteria allowed the identification of 12 papers potentially suitable for the systematic review (Figure 1).

## 3. Results and Discussion

The initial search resulted in a total of 114 papers found in the three databases. After applying the exclusion criteria and full reading of the remaining articles, 12 papers were considered eligible and included in the present section. These 12 studies are summarized in Table 1.

Most of the articles are from Spain (*n* = 9), two from Portugal and one from France. Eleven studies detected BGAV in birds, and only one refers to ungulate mammals [27]. In the class Aves (birds), the most represented order is Galliformes and the most represented family is Phasianidae. The Red-legged Partridge (*Alectoris rufa*) and Common Pheasant (*Phasianus colchicus*) are described in nine and five studies, respectively. Apparently, according to reports to date, this virus affects phasianids in particular, of which some species are considered game animals all over the world [32]. BAGV has been reported in other pheasant species, i.e., the Himalayan Monal Pheasants (*Lophophorus impejanus*) [13]. Another study describes experimental infection in Grey Partridges (*Perdix perdix*), showing that this species is also susceptible to neurological diseases induced by BGAV [33]. Other birds from different orders which were proven to be infected with BAGV in one paper each were the Corn Bunting (*Emberiza calandra*) [14], Cinereous vulture (*Aegypius monachus*) [31], Common Woodpigeon (*Columba palumbus*) [12], Eurasian Magpie (*Pica pica*) [2], Eurasian Spoonbill (*Platalea leucorodia*), Green woodpecker (*Picus viridis*), and White Stork (*Ciconia ciconia*) [31]. Experimental infection in House Sparrows (*Passer domesticus*) was not able to induce mortality or signs of disease in this species, and no viremia was detected during the experimental period [34]. Apparently, apart from game birds, there is no predisposition towards any other family or genus of birds [34]. However, the number of positives reported is too small to draw a conclusion in this respect. On the contrary, we may assume that certain behaviors that encourage contact with mosquito areas may increase the risk of infection.

In regard to clinical signs induced by BAGV infection, severe weight loss was the main finding across infected animals, and also apathy, weakness, reluctance to move, and neurological symptoms including paralysis, disorientation, ataxia, and unresponsiveness (circling and twisted neck were also observed) [25,33,34]. Besides the central nervous system, BAGV appears to have a tropism for endothelial cells and causes a severe hemolytic process [35]. In the most affected countries, like Portugal and Spain, veterinarians should not underestimate these non-specific clinical signs and start including BAGV infection in the list of differential diagnoses, namely in birds with hemolytic anemia of unknown origin.

Mortality values as high as 30–40% have been described, between 6 and 10 days post-infection (dpi). Viremia was observed at 3–5 dpi, with a peak viral RNA detection in blood circulation at 3 dpi in an experimental trial. BAGV-neutralizing antibodies were detected in all partridges by 7 dpi [33,34]. Virus shedding occurs from 3 dpi onwards (until 11 dpi), either through cloacal or oral routes. Oropharyngeal swabs proved to be more suitable for a consistent detection of the virus. On the other hand, feathers appear to be the preferred option to detect the viral RNA after long periods, when BAGV is not already present in blood or cloacal and oropharyngeal swabs [34]. In the case of dead animals, brain tissue is considered an organ of choice for detecting BAGV [13]. In some studies in which several organs were screened independently, all or most of the samples tested positive in brain tissue [11,12,24].

Both the Japanese encephalitis (JE) and Ntaya groups from the genus *Orthoflavivirus* comprise a number of pathogens associated with neurological diseases in different vertebrate species. As they are both maintained in nature in a cycle involving avian reservoir hosts and *Culex* spp. as vectors, the possibility of them circulating simultaneously is high [36,37]. From the JE group, WNV and USUV stand out in Southwestern Europe. Infection with both of these viruses and BGAV produces antibodies that cross-react in serological tests, leading to possible false-positive reactions [38]. Enzyme-linked immunosorbent assay (ELISA) can be used as a first-line serological test, but viral neutralization test (VNT) is more specific and therefore used as confirmatory, although it still exhibits some cross-reactivity, especially between agents from the same group like WNV and USUV [38,39]. Regarding molecular diagnosis of flaviviral infections, it relies more on generic RT-PCR assays. The generic detection of viral species of both serocomplexes in a single test may provide more accurate and rapid diagnostic results in monitoring programs, and a reliable, specific, and highly sensitive tool for rapid detection and differentiation of JE and Ntaya *Orthoflavivirus* (quantitative duplex qRT-PCR) has been described as helpful to be used as a screening tool in routine avian surveillance [37]. Another effective protocol for BAGV detection that has been described is the qRT-PCR method targeting the Bagaza NS5 gene [24]. It would be very interesting to be able to determine the exact prevalence of each *Orthoflavivirus* in the area under study every now and then, under a surveillance system regulated by the respective national authorities.

The full-length genome sequence of BAGV has already been obtained a long time ago [40]. The isolated sequence of BAGV from the first time in Spain showed greater similarity with the strain from Africa than with the strain from India [11]. Migratory birds may have contributed to the transfer of BAGV between continents, just like with other flaviviruses, but this theory has not yet been confirmed [41,42]. More recently, an in-depth analysis of the BAGV sequence isolated from Portugal revealed a phylogenetic proximity to the BAGV from Spain, indicating that the virus has most likely arrived via the cross-border route [4]. This might have a major impact in terms of ecological conservation since there are species with threatened conservation status that are endemic to the Iberian Peninsula. Any threat shared between the two countries must be taken into account and tackled together since the genetic heritage of species with few living individuals in the wild is very significant.

In conclusion, the Red-legged Partridge is considered a suitable target species for BAGV surveillance, and more extensive epidemiological research in susceptible birds is needed in order to control the introduction, maintenance, and spread of the disease. More active surveillance studies could be carried out, with large-scale sampling of birds caught in the wild. BAGV is potentially a growing threat to the wild birds of the Iberian Peninsula, considered a high-risk area, and prevention is the best approach to mitigate that threat.

## Figures and Tables

**Figure 1 vetsci-12-00113-f001:**
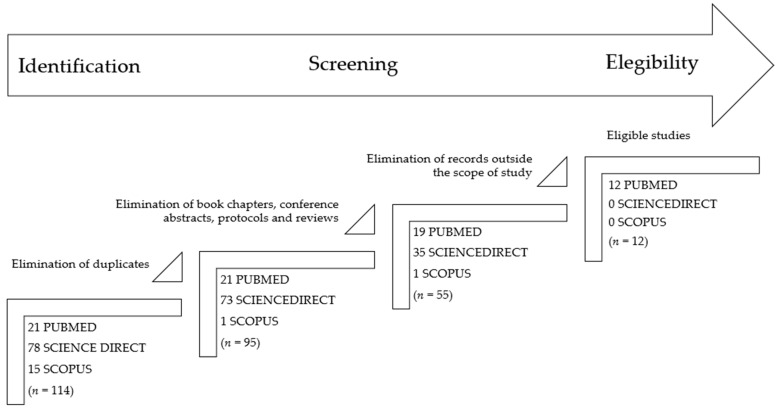
PRISMA flowchart describing the steps of the record selection procedure and inclusion/exclusion criteria.

**Table 1 vetsci-12-00113-t001:** A summary of the studies reporting evidence of BAGV circulation in Europe (by year of publication).

Location	Sampling Date	Sample Type	Diagnostic Assay	Number of Positive/Total (%)	Sequencing (GenBank ID)	Species Identified with BAGV	Reference
Cadiz, Andalusia—Spain	September 2010	Tissues (heart, intestine, lung, liver, kidney, brain, and feathers)	RT-PCR (NS5 gene segment; segment 214 bp) Virus isolation (AF, CM, ECE, VS)	13/13 (100)	HQ644143 HQ644144	*Alectoris rufa; Phasianus colchicus*	[11]
Cadiz, Andalusia—Spain	September 2010 2010–2011	Tissues (heart, intestine, lung, liver, kidney, brain, and feathers)	qRT-PCR (NS5 gene)	11/11 (100) 0/81 (0)		*Alectoris rufa*	[24] ^1^
Southwestern Spain	August 2010	Oropharyngeal and cloacal swabs Tissues (brain, oral mucosa, pectoral muscle, trachea, lung, heart, liver, spleen, pancreas, duodenum, caecal tonsils, kidney, bursa of Fabricius, thymus and skin with feather follicles)	qRT-PCR Immunohistochemistry	13/13 (100)	AY632545.2	*Alectoris rufa; Columba palumbus; Phasianus colchicus*	[12]
Cadiz, Andalusia—Spain	August–October 2010	Blood Tissue samples Oropharyngeal and cloacal swabs	RT-PCR	11/14 (78.6)		*Alectoris rufa; Phasianus colchicus*	[25]
Cadiz, Andalusia—Spain	October 2011–February 2012	Serum Brain	VNT RT-PCR	25/172 (14.5) 0/172 (0)		*Alectoris rufa; Phasianus colchicus*	[26]
France	September 2009–February 2010	Blood	VNT	0/73 (0)		*Capreolus capreolus; Sus scrofa*	[27]
Extremadura—Spain	October 2017–December 2019	Tissues (blood, brain, heart, intestine, liver, lung, muscle, kidney, spleen, stomach, pancreas and the pulp of immature feathers)	VNT	0/157 (0)		*---*	[28]
Serpa, Alentejo—Portugal	September 2021	Tissues (feather pulp, brain, heart, kidney, spleen, and intestine) Growing feathers (live birds)	qRT-PCR (NS2b, NS5, and 3′ NT region)	9/12 (75) 4/30 (13.3)		*Alectoris rufa; Emberiza calandra*	[14]
Cadiz, Andalusia—Spain	October 2019	Growing feathers, heart, brain, liver, spleen and kidney; Tissues (heart, brain, spleen, liver, kidney, lung, skeletal muscle, skin, cecal tonsils, adrenal glands, gonads and pancreas	qRT-PCR (NS5 gene segment; partial segment 222 bp) Immunohistochemistry	4/4 (100)	OK424741 OK424742	*Alectoris rufa*	[29]
Cadiz, Andalusia—Spain	October 2019–August 2021	Brain	qRT-PCR (NS5 gene segment) Virus isolation (Vero and BSR cells)	4/4 (100)	PP236854 PP236853 PP236852 PP236851	*Alectoris rufa*	[30]
Mértola, Alentejo—Portugal	September 2023	Tissues (kidney, spleen, heart and feather follicles)	qRT-PCR (NS5 gene; 342 bp region within the NS1 gene)	4/7 (57.1)	PP130723	*Pica pica*	[2]
Cadiz and Seville, Andalusia—Spain	July 2021–February 2022 January–December 2021	Tissues (brain, growing feathers) Oropharyngeal and cloacal swabs	RT-PCR (NS5 gene)	32/89 (35.9) 4/215 (1.9)	PP887449 PP887448 PP887447 PP887446 PP887445 LC730845	*Alectoris rufa*; *Phasianus colchicus* *Picus viridis*; *Platalea leucorodia*; *Ciconia ciconia*; *Aegypius monachus*	[31]

AF—allantoic fluid; CM—chorioallantoic fluid; ECE—embryonated chicken egg; NS—non-structural; NT—nontranslated; qRT-PCR—quantitative (real time) reverse transcription polymerase chain reaction; VNT—viral neutralization test; vs.—viscera. ^1^ Methodological study, does not report the presence of the virus, but rather test design.

## Data Availability

No new data were created.

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
