# Peer review of "Detection of Bagaza Virus in Europe: A Scoping Review"

_vetsci, 2025, doi:10.3390/vetsci12020113_

Round 1
Reviewer 1 Report
Comments and Suggestions for Authors
As Orthoflavivirus (Ntaya serocomplex), BAGV is maintained through an epidemiological 50 cycle involving mosquitoes as competent vectors and birds as natural reservoirs and virus 51 amplifier. In this sbmissiom, the authors reviewed all available recent information on BAGV detection in Europe.
In general, the paper is interesting study, although the comparison between BAGV and other flaviviruses in the Introduction section needs to be strengthened.
In the Discussion section, most of the content merely lists the research progress, lacking in - depth analysis..
Author Response
The authors greatly appreciate the time and effort you have put into providing valuable feedback on the manuscript.
Comment 1:In general, the paper is interesting study, although the comparison between BAGV and other flaviviruses in the Introduction section needs to be strengthened.
Response 1: Thank you very much. More topics about BAGV and other orthoflaviviruses have been added to the introduction section.
Comment 2: In the Discussion section, most of the content merely lists the research progress, lacking in - depth analysis..
Response 2: Thank you. We have enriched the discussion.
Reviewer 2 Report
Comments and Suggestions for Authors
Thank you for the opportunity to review the article entitled “Detection of Bagaza virus in Europe: a scoping review”. Overall, this is very interesting paper, the topic is appropriate and thematically good. I like reading it.
Briefly, the paper is a review on Bagaza virus (BAGV) an emerged virus in Europe. The natural transmission cycle of this virus is perpetuated by Culex spp. mosquitoes and viraemic birds. The BAGV cause infection in several game birds from the family Phasianidae and is antigenically similar to other flaviviruses from the Japanese encephalitis serocomplex, such as the West Nile and Usutu viruses. Severe implications in animal health have already been described, but some aspects of the dynamics of transmission and the limits of zoonotic potential of BAGV still need to be clarified. The presented study is a systematic review of the BAGV reports in Europe published in main databases such as Pubmed, ScienceDirect and Scopus.
Minor concerns are listed below.
In Table 1 - please ad columns on status of BAGV positive birds (live, dead, experimental, …) and change wording “sampling location” in “location”. Since is not clear if this is “real location of birds found dead, or found ill” or this is “the sampling location - like location of the lab where the samples were taken, …”
Author Response
The authors greatly appreciate the time and effort you have put into providing valuable feedback on the manuscript.
Comment 1: In Table 1 - please ad columns on status of BAGV positive birds (live, dead, experimental, …)
Response 1: We have decided not to include that information in a new column because all the positive birds were found dead, with the exception of “an electrocuted white stork (Ciconia ciconia) from Malaga province and a cynereous vulture (Aegypius monachus) in poor body condition admitted to a wildlife rescue centre (Granada province)”. These 2 cases were reported by Gonzálvez et al. (2024), and the paper does not inform on whether the birds have died in the meantime or not.
Comment 2: In Table 1 (…) change wording “sampling location” in “location”. Since is not clear if this is “real location of birds found dead, or found ill” or this is “the sampling location - like location of the lab where the samples were taken, …”
Response 2: Thank you for noticing this. Not all studies refer to location in the same way. We have made the change to “location”.
Round 2
Reviewer 1 Report
Comments and Suggestions for Authors
Can be accepted.